# Sediment source and dose influence the larval performance of the threatened coral *Orbicella faveolata*

Xaymara M. Serrano[1,2¤]*, Stephanie M. Rosales[1,2], Margaret W. Miller[3], Ana M. Palacio-Castro[1,2], Olivia M. Williamson[4], Andrea Gomez[5], Andrew C. Baker[4]*

1 Cooperative Institute of Marine and Atmospheric Studies, Rosenstiel School of Marine, Atmospheric, and Earth Science, University of Miami, Miami, Florida, United States of America, 2 Atlantic and Oceanographic Meteorological Laboratory, National Oceanic and Atmospheric Administration, Miami, Florida, United States of America, 3 SECORE International, Miami, Florida, United States of America, 4 Department of Marine Biology and Ecology, Rosenstiel School of Marine, Atmospheric, and Earth Science, University of Miami, Miami, Florida, United States of America, 5 National Oceanic and Atmospheric Administration, Greater Atlantic Regional Fisheries Office, Gloucester, Massachusetts, United States of America

¤ Current address: Habitat Conservation Division, National Marine Fisheries Service, West Palm Beach, Florida, United States of America
* xaymara.serrano@noaa.gov (XMS); abaker@earth.miami.edu (ACB)

**Data Availability Statement:** All data and code for the larval survivorship, settlement and oxygen consumption analysis can be found in Zenodo at https://doi.org/10.5281/zenodo.8022069. Raw

## Abstract

The effects of turbidity and sedimentation stress on early life stages of corals are poorly understood, particularly in Atlantic species. Dredging operations, beach nourishment, and other coastal construction activities can increase sedimentation and turbidity in nearby coral reef habitats and have the potential to negatively affect coral larval development and metamorphosis, reducing sexual reproduction success. In this study, we investigated the performance of larvae of the threatened Caribbean coral species *Orbicella faveolata* exposed to suspended sediments collected from a reef site in southeast Florida recently impacted by dredging (Port of Miami), and compared it to the performance of larvae exposed to sediments collected from the offshore, natal reef of the parent colonies. In a laboratory experiment, we tested whether low and high doses of each of these sediment types affected the survival, settlement, and respiration of coral larvae compared to a no-sediment control treatment. In addition, we analyzed the sediments used in the experiments with 16S rRNA gene amplicon sequencing to assess differences in the microbial communities present in the Port versus Reef sediments, and their potential impact on coral performance. Overall, only *O. faveolata* larvae exposed to the high-dose Port sediment treatment had significantly lower survival rates compared to the control treatment, suggesting an initial tolerance to elevated suspended sediments. However, significantly lower settlement rates were observed in both Port treatments (low- and high-dose) compared to the control treatment one week after exposure, suggesting strong latent effects. Sediments collected near the Port also contained different microbial communities than Reef sediments, and higher relative abundances of the bacteria Desulfobacterales, which has been associated with coral disease. We hypothesize that differences in microbial communities between the two sediments may be a contributing factor in explaining the observed differences in larval performance. Together, these results suggest that the settlement success and survival of *O. faveolata*

reads and metadata for the microbial 16S of sediment samples are available in NCBI's SRA under BioProject ID PRJNA1008169. All data and code for the sediment microbial analysis can be found at https://github.com/srosales712/Larvae_Sediment.

**Funding:** This research was supported with funds from MOTE (Protect Our Reefs grants POR-2015-15 and POR-2016-16) awarded to X. M. Serrano, A. C. Baker and M. W. Miller, and funds from NOAA's Coral Reef Conservation Program (Project ID 31147) awarded to X. M. Serrano and M. W. Miller. The funders had no role in study design, data collection and analysis, decision to publish, or preparation of the manuscript.

**Competing interests:** The authors have declared that no competing interests exist.

larvae are more readily compromised by encountering port inlet sediments compared to reef sediments, with potentially important consequences for the recruitment success of this species in affected areas.

## Introduction

Dredging and port construction activities can induce chronic sedimentation and turbidity stress in surrounding coral reef habitats [1–6]. Sedimentation caused by dredging is often more harmful to corals than sedimentation caused by other activities, due to its acute onset and often long duration, typically lasting many months to years [1,7]. Sedimentation effects vary depending on grain size and composition (e.g., [7–12]), and dredging activities can release large quantities of sediments in the water column that are typically of finer grain size compared to naturally occurring sediments (<63 μm, [7]). These finer sediments tend to attenuate more light and are more prone to being resuspended, thus causing a greater net reduction in light essential for photosynthesis [10]. Finer sediments also have an adhesive, clay-like texture that is more resistant to bioturbation when deposited [13], can lead to a change in the bacterial community [12], and is more likely to become anoxic [8,14]. Finally, suspended sediment plumes, especially those with finer grain size, can travel several km away from the dredging site (e.g., [9,15,16]), and can also release contaminants [17–19] and pathogens [11,20,21] which may potentially transmit coral diseases such as Stony Coral Tissue Loss Disease [22].

Sediments can negatively affect corals throughout their life cycle, suppressing coral health, condition, and survival via multiple mechanisms (reviewed in [7,12,13,23]). Reported effects on early life stages of corals include reductions in the fecundity of parent colonies [24], fertilization [25–28], larval settlement [25,29–33], and recruit survival [5,29,34,35]. Tolerance to sedimentation exposure is estimated to be an order of magnitude lower for coral recruits compared to adults [13,23,36], although most of this work has been conducted on Pacific coral species (reviewed in [12,13,23]). Even a thin layer of sediments not harmful to adult corals can be detrimental to coral larvae [23,25,30,33,37,38], both by physically obstructing settlement surfaces [29,32], and impairing biotic settlement cues such as those from crustose coralline algae (CCA) [32]. However, the mechanisms and cues by which sediments can affect settlement and recruitment success, or the benthic habitat quality (e.g., [33]) are poorly understood. Recent work suggests that coral larvae prefer distinct microbial and/or chemical signature types in certain types of CCA [39–42], suggesting that specific bacterial communities can serve as environmental cues for coral settlement [39,41]. How and whether microbial communities present in sediments can negatively affect the ability of coral larvae to identify these cues remains an open question.

Whereas most of the work on the effects of sedimentation on coral recruitment has focused on the benthos (i.e., sediment-covered surfaces), the effects of suspended sediments on coral larval survival and settlement remain poorly understood. For the few Pacific species studied to date, findings suggest that coral larvae may be generally resistant to brief exposures of elevated suspended sediment concentrations as high as 100–1000 mg/L ([26,43] but see [25]). However, little is known regarding the effects of suspended sediments on Caribbean coral larvae and recruits, and this information is critical for developing appropriate conservation plans for these species. Poor understanding of coral responses to sediment disturbances during their most susceptible stages can result in inappropriate management of dredging projects,

preventable coral losses, and unnecessarily high costs to adaptively manage operations (corrective actions) and provide compensatory mitigation to offset impacts.

The aim of the present study was to investigate the performance of coral larvae of an important Caribbean reef-building species (*Orbicella faveolata*) following a short-term (pulse) suspended sediment exposure. *O. faveolata* (Ellis and Solander, 1786) is listed as threatened under the U.S. Endangered Species Act [44] but is present in southeast Florida in the vicinity of two large ports with expansions (deepening/widening) planned in the near future (Port Everglades and Port of Miami). The most recent large-scale deepening project at Port of Miami (which occurred from 2013–2015) resulted in severe sedimentation impacts to local coral reef habitats [1,3,15,45], although the severity of those impacts has been disputed [46]. In addition, recent work exposed adult *O. faveolata* corals to Port of Miami sediments during a 96-hr period and showed adverse effects on tissue regeneration capacity compared to no sediment controls at concentrations as low as 50 mg/L (~4 NTU, [47]). In this study, we exposed *O. faveolata* larvae to suspended sediments collected near the Port of Miami channel and compared their performance with larvae exposed to reef sediments collected from the natal reef of the parent colonies in the Florida Keys. We compared the effects of the two different sediment types on the survival, settlement, and respiration of coral larvae, and also characterized the bacterial communities of these sediments using 16S rRNA gene amplicon sequencing to assess the potential role of microbial communities in affecting coral performance. To our knowledge, this is the first study to investigate the effects of suspended sediments on larvae of western Atlantic (Caribbean) coral species, and one of few studies to use "wet" sediments that have not been dried or sterilized prior to exposure, which may have masked some of the biological or chemical effects of other sedimentation studies reported to date.

## Methods

### Coral larvae collection

*O. faveolata* larvae were collected and cultured as described in [48,49]. Gamete 'egg-sperm' bundles were collected seven nights after the full moon (night of 14[th] August 2017) from colonies that spawned at two reefs in the upper Florida Keys (Horseshoe and Sand Island reefs). Once the bundles broke apart, equal volumes of gametes from 3–4 parental genets were combined for fertilization. Resulting embryos were maintained in water collected from their natal reef at a land-based facility in the Keys (either in static bins with regular water changes or in a recirculating system with mesh-floored bins and a constant drip input) and cultured for ~2 days so that unfertilized eggs would disintegrate, leaving behind only developing larvae. Embryos were kept in a shaded area under ambient light conditions and a mean temperature of 29°C. Larvae were then transferred in bins with lids and aeration to the University of Miami for experiments.

### Sediment collections and maintenance

Immediately prior to starting the larval experiments (24–48 h), two batches of sediments were collected, one from a "Reef" origin and another from a "Port" origin. Reef sediments were collected at Horseshoe Reef in Key Largo, from the same site where the coral parent colonies occurred (25°8'19"N, 80°17'41"W, Fig 1). Port sediments were collected from a site located ~200 m north of the recently-dredged channel of Port of Miami (25°45'50"N, 80°5'56"W, Fig 1), reported in [3] as having detrimental effects from dredging-induced sedimentation on resident adult corals. This site was shown by [3] as the most severely impacted location surveyed, with the highest sediment cover (43%) and highest prevalence of sediment "halos" (indicating partial mortality) compared to the reference stations.

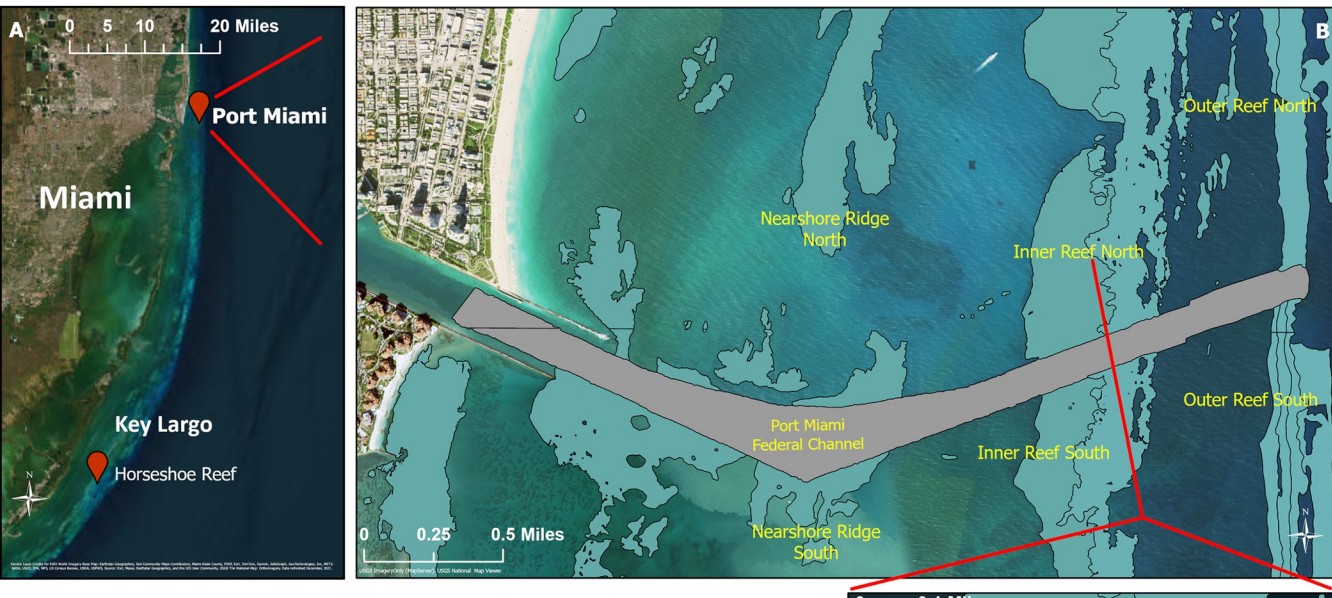

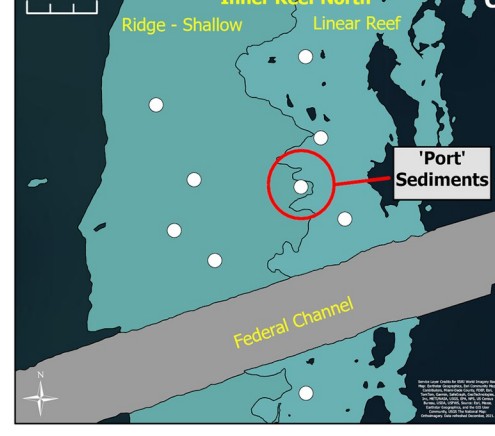

**Fig 1.** (A) Map of locations on the coast of southeastern Florida from which Port and Reef sediments were collected (near Port of Miami and Horseshoe Reef in Key Largo). (B) Map of coral reef habitats around the Port of Miami federal channel. (C) Port sediments were collected from a site within the Inner Reef located ~200 m north of the recently dredged channel of Port of Miami (circled in red), which was the most severely impacted location surveyed due to sedimentation compared to the reference stations by [3]. Copyright Information: Fig 1A and 1C use the World Imagery Base Map from ESRI. Fig 1B uses the USGS ImageryOnly Base Map from the USGS National Map Viewer. All maps were produced using ArcPro 3.1.0. Service Layer Credits for ESRI World Imagery Base Map: Earthstar Geographics, Esri Community Maps Contributors, Miami-Dade County, FDEP, Esri, TomTom, Garmin, SafeGraph, GeoTeochnologies, Inc, METI/NASA, USGS, EPA, NPS, US Census Bureau, USDA, USFWS, Source: Esri, Maxar, Earthstar Geographics, and the GIS User Community, USGS The National Map: Orthoimagery. Data refreshed December, 2021.

All sediments were taken from the upper ~3 cm of the sediment layer (surficial layer) by sliding a metal tool below the sediment-water interface and then scooping them into a certi-fied-clean HPDE sample container. Three to four containers were collected per site and com-bined to produce a single "batch" for experiments. Sediments were then brought to the laboratory, homogenized in a beaker of seawater and kept in continuous movement and aera-tion until used in experiments. Prior to use, sediments were allowed to settle and water was siphoned out with care to avoid disturbing the fine particles on the surface. Sediments were not dried or sterilized to avoid changing their biological or chemical properties and be able to assess any physical (e.g., abrasion) or biological (e.g., pathogenic) stress effects. Hence, a subset of the sediments was analyzed for dry weight (ratio to wet weight) after drying the sediments

thoroughly in an oven. Then, for setting up the experimental treatment doses, sediments were wet-weighed (after adjusting for the percentage of moisture content) to avoid killing any microorganisms present while retaining as many fine particles as possible.

A subset of the dried sediments was analyzed for grain size by sieving and pipetting. Grain size limits were defined according to the Udden-Wentworth US standard classification [50], while statistical parameters were determined using graphical measures from the cumulative size distributions as shown in S1 Fig. A summary of these parameters are provided in S1 Table. Reef sediments can be described as a well-sorted medium sand with virtually no silt or gravel. Conversely, Port sediments can be described as a poorly-sorted, medium sand with small amounts of gravel (approx. 1.97% >2000 μm) and silt and clay-sized particles (also referred as "mud", approx. 1.25% <63 μm). No chemical or organic matter analysis of the sediments were conducted.

## Experimental setup

Fifty 8-day-old larvae were added to individual glass vials containing 45 mL filtered seawater (5 μm). Vials were allocated randomly to one of five different treatments: (1) a control treatment (no sediment), (2) a low "Reef" suspended sediment treatment (20 mg/L of homogenized sediment from Horseshoe Reef added), (3) a low "Port" sediment treatment (20 mg/L sediment from the Port site added), (4) a high "Reef" sediment treatment (100 mg/L sediment from Horseshoe reef added), and (5) a high "Port" sediment treatment (100 mg/L sediment from the Port site added). Nine to ten replicates were used per treatment. Vials were attached to a titer plate shaker set to 300 rpm to continuously maintain sediments in suspension for 24 h. These treatment parameters (duration and shaker setting) were determined in initial trials to assess the best exposure times that caused minimal mortality in control larvae and maintained both fine and coarse sediment particles in continuous suspension.

Sediment concentration treatments were chosen to correspond to sub-lethal and lethal responses to suspended sediment concentrations for "tolerant" versus "sensitive" species as described by [12]. In addition, our goal was to test a low concentration of suspended sediment (20 mg/L) representing typical conditions near dredging operations. Jones et al. [13] analyzed suspended sediment concentrations over a 30-day running mean period during the Barrow Island dredging project in Australia and found that the 95th percentile was 21 mg/L for 7 sites located at distances <1 km from the dredging. This is about twice the concentrations typically observed in natural reefs in southeast Florida (~10 mg/L; [51]), but within the range observed in inshore reefs in Hawaii ([52]), and in turbid reefs in the Great Barrier Reef, which often exceed 50 mg/L during natural wind-wave events ([53–55]). In addition, although our 100 mg/L (high-dose) suspended sediment treatments are approximately 5 times higher than those typically observed during dredging operations, such conditions do occur over short periods (see [53]).

## 24-hour treatment exposure: Larval survival assay

After 24 h of exposure, swimming larvae remaining in each vial were pipetted out, counted and examined with a microscope to determine the proportion that had survived. The vial was also carefully examined for potential settlers. A subset of larvae (N = 6 per vial) was used to assess changes in respiration rates as described below. Remaining larvae were allowed to recover for one week in filtered seawater (without sediments) in the same vials before being transported to the Florida Keys to conduct the settlement assays.

## 24 hour and 1-week assessment of coral larval respiration

Respiration was measured using a microplate reader developed by Loligo Systems (Denmark), which allows measuring real-time respiration rates in each one of 24 wells using an optical fluorescence oxygen sensing technology (SDR SensorDish Reader®, PreSens, Germany) as described in [56–58]. This system comes pre-calibrated specifically for the reader used, and oxygen solubility is continuously calculated with oxygen sensors (optrodes) attached to the bottom of each well. Larval respiration rates were assessed at two time points, immediately after the 24 h treatment exposure and 1-week post-exposure. At each time point, larvae (N = 6) were taken at random from each vial and placed in each of the wells of a sealed glass 125-μL microplate (without any sediments). A total of 8–10 replicate wells were used per treatment. Four additional control wells were used in each plate without any larvae to measure any background respiration in the treatment water.

Larvae were visually inspected for normal swimming behavior prior to each run. After each plate was prepared, wells were inspected to ensure there were no air bubbles, and an oxygen-impermeable seal was created using a silicone membrane covered with parafilm and a compression block. To maintain constant temperature, the microplate was placed inside a temperature-controlled flow-through water bath (included as part of the system), in a light-controlled room. The plate was then placed in a titer shaker at a low speed to maintain constant water motion. Finally, measurements of oxygen concentrations were taken every 15 seconds throughout the run and recorded using the SDR v 4.0.0 software (PreSens, Germany). Oxygen concentration was plotted as a function of time for individual wells and the first 10–20% ($pO_2$ in % air saturation) linear decreases in oxygen were used to calculate the oxygen consumption rate (an example plot of oxygen consumption over time for 6 *O. faveolata* larvae placed in a single well of the microplate is shown in S2 Fig). Any portion of the slope that dipped below 70% air saturation was not used for analysis. Respiration rates were then corrected for background respiration (i.e., in control wells) and calculated as nanomoles of oxygen consumed per larva per minute.

## 1-week post-exposure: Larval settlement assays

Settlement assays were conducted by pooling all the larvae remaining in each treatment and then redistributing N = 15 larvae to individual vials containing filtered seawater and a piece of rubble collected from Horseshoe reef to act as settlement cue. A total of 10 replicate glass vials per treatment were used to quantify larval settlement per treatment. After 24 h, the total number of larvae still swimming, or that had settled on the rubble, was manually scored using fluorescence microscopy. Larvae were classified as "settlers" only if they displayed visible signs of settlement (attachment to the substrate) and metamorphosis (i.e., transition from pear-shaped to flat/disc shape). To determine the cumulative percentage of larval settlement, the total number of settled corals (spat) was divided by the initial number of larvae added to each container.

## 16S rRNA gene sequencing of sediment samples

Immediately upon collection, ~1 gram from each of the Port and Reef sediment sample containers collected per site was preserved in RNAlater for 16S sequencing. The DNA was extracted and PCR amplified with 16S rRNA gene V4 primers [59] with the Platinum Hot Start PCR Master Mix (2X) (ThermoFisher Scientific, Waltham, MA). The master mix was a 48-μl reaction and 2 μl of DNA template was used. The DNA was then amplified as follows: 94 ˚C for 3 minutes (1X), 94 ˚C for 45 seconds (35X), 50 ˚C for 60 seconds (35X), 72 ˚C for 90 seconds (35X), and 72 ˚C for 10 minutes (1X). Amplified products were cleaned using AMPure XP beads (Beckman Coulter, Brea, CA), normalized to 4 nM, and 5 μl of each normalized

sample was pooled. The samples were sequenced on a MiSeq with PE-300v3 kits at the Halmos College of Natural Sciences and Oceanography at Nova Southeastern University.

## Statistical analysis

**Larval performance.** All analyses of larval performance data were done in R v.4.1.2. The effects of the sediment treatments on the proportion of larval survivorship and settlement were analyzed using binomial generalized linear mixed models (GLMM) with the lme4 package [60]. The models included sediment treatment as a fixed effect and replicate (vial) as a random effect. Model overdispersion was tested with the package performance (v0.10.9) and when present was corrected by introducing observation-level random effects [61]. Models with and without random effects were compared and the best model was chosen to minimize the Akaike information criterion (AIC). Model predictions (survivorship and settlement proportions as well as the odds ratio between the control and the treatments) were plotted using the plot_model function from the sjPlot package [62] including 95% confidence intervals. Pairwise differences between control and sediment treatments were evaluated with the dunnettx method and an alpha value of 0.05 using the emmeans package (v.1.8.3) [63]. Larval respiration rates after 24 h of treatment exposure and 1-week post-exposure were analyzed using a linear mixed-effects model (LMEM) with the lme4 package. The model included sediment treatment and time point (exposure and recovery) as fixed interactive effects, and vial, plate, and well as random effects. Model selection was performed with stepwise backward elimination of non-significant terms using the "step" function of the lmerTest R package [60]. Posthoc differences among significant factors were evaluated with the emmeans package. All data and code for larva survivorship, settlement and oxygen consumption can be found at Zenodo [64].

**Microbiome analysis of sediment samples.** DNA sequences were demultiplexed and further processed with Qiime2-2022.11 [65]. Since the reverse reads were of poor quality only the forward reads were analyzed. Using the program Cutadapt the primers were removed [66], and Amplicon Sequence Variants (ASVs) were generated with the program DADA2 [67] trimmed at the 10 and 205 bp positions. The ASVs were taxonomically classified with the classify-consensus-vsearch function and the silva 138 99 database [68]. If the sequences were annotated as chloroplast or mitochondria they were removed from the analysis. The data were then uploaded to R and converted into a Phyloseq v1.26.1 data object for analysis.

Alpha-diversity was evaluated by rarefying to a minimum sequence depth of 60,000. ASVs that had zero across all samples were removed. Three alpha-diversity metrics, richness (i.e., observed), Shannon-Wiener, and the Inverse Simpson index, were generated with the phyloseq estimate_richness function. The respective values were then tested for normality using qqnorm, qqline, and the Shapiro test. Upon passing normality, significance between sites was tested with an Analysis of Variance (ANOVA).

For beta-diversity, ASVs were filtered if they summed to <10 in 30% of the data. The filtered data were then transformed to centered log ratios (CLR) using the package Microbiome v0.9.99. The Vegan v2.5.4 package was used for beta-diversity testing for both within and between group differences. Dispersion of the samples (within group) was tested by using the function Vegdist (method = "Euclidean", Permutations = 999) and Betadisper and was tested for significance with a permutation test. A permutational multivariate analysis of variance (PERMANOVA, function Adonis, method = "Euclidean", Permutations = 999) was used to evaluate significance between groups.

To identify differentially abundant microbial ASVs between sites, the program Analysis of Compositions of Microbiomes with Bias Correction (ANCOM_BC) was used [69].

ANCOM_BC was set with an alpha <0.001 and a W statistic >90. The ASVs were further evaluated if they had a log-fold change < -3.5 or > 3.5.

## Results

### Larval performance

Suspended sediment treatments, especially from the Port source, significantly impacted the performance of *O. faveolata* larvae (Figs 2 and S3; S2 and S3 Tables). Survival proportions after 24 h in the Control and Low Reef treatments were above 0.94, but exposure to Low Port, High Reef, and High Port reduced survival proportions to 0.91, 0.88, and 0.82, respectively. However, only the High Port treatment had significantly lower survivorship rates compared to the Control treatment (Dunnett pairwise test p = 0.009; Fig 2A), and significantly lower odds ratios (0.26; S3 Fig). Larval settlement proportions 1-week after treatments were 0.21–0.22 in the Control and Low Reef treatments, but were reduced to less than 0.12 in the High Reef, 0.09 in the Low Port, and 0.10 in the High Port treatments. Compared to Controls, only the Low and High Port treatments had significantly lower settlement rates (Dunnett pairwise test p = 0.02 and 0.03, respectively; Fig 2B). The odds ratios for settlement were <0.5 for the larvae in the High Reef treatment, and <0.4 for the larvae in Low and High Port treatments compared to the Control (S3 Fig).

None of the sediment treatments affected the larval respiration rates at either of the time points (S4 Table, S4 Fig). Average respiration rates per larva were $9.33 \times 10^{-4} \pm 2.53 \times 10^{-4}$ nmol $O_2$ min$^{-1}$ (mean ± SD) across treatments after 24 h of exposure and declined to $6.01 \times 10^{-4} \pm 1.97 \times 10^{-4}$ nmol $O_2$ min$^{-1}$ after a week of recovery, but this was only marginally significant (p<0.06, df = 1, F = 15.83, S4 Table).

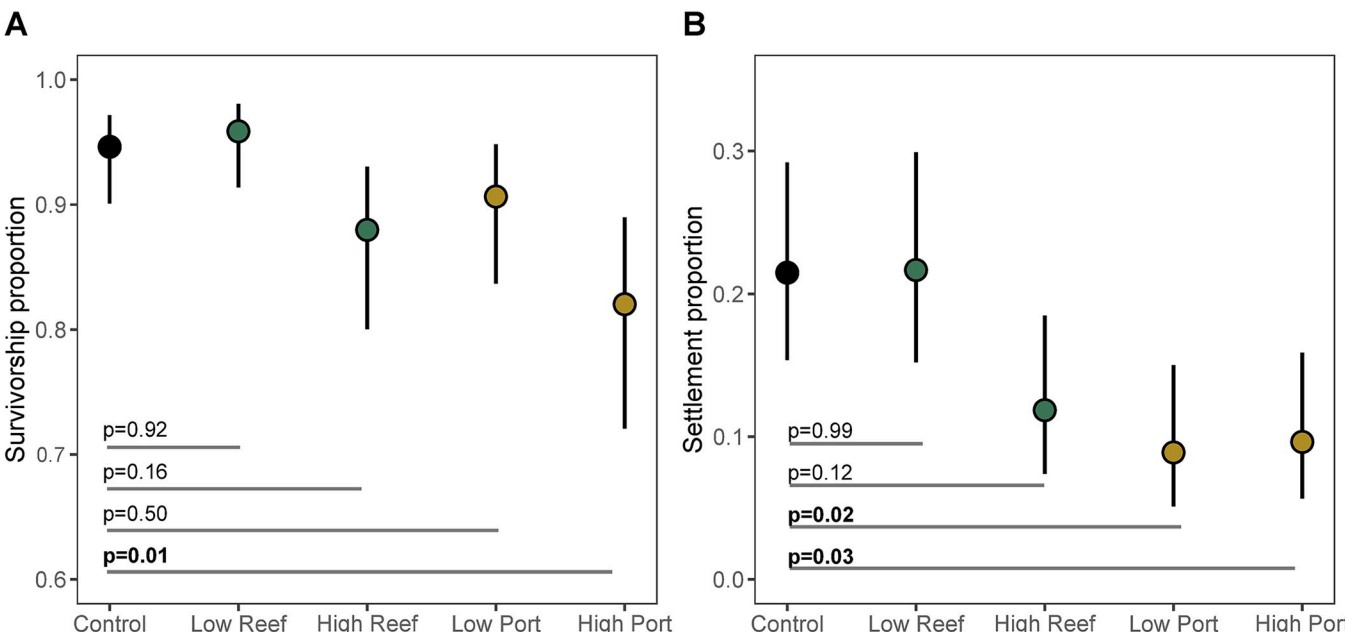

**Fig 2. Effects of sediments on *Orbicella faveolata* larvae (model estimated values ± 95% CI).** (A) Proportion of larval survival after a 24 h exposure to experimental treatments. (B) Proportion of larval settlement after one week of recovery from experimental treatments. P-values for specific pairwise comparisons are shown in each panel and were calculated using the dunnettx method for 4 tests. Colors match Figs 3 and 4 and denote either Port or Reef sediment treatments.

## Microbiome analysis

A total of seven sediment samples (Port N = 3 and Reef N = 4) resulted in sequence counts between 93,626 and 419,403 (median = 171,281). After processing and filtering, 1,852 ASVs were used for the analysis. Alpha-diversity was not significant between sediment types, but was slightly higher in the Reef sediments (richness: Reef = 3604 vs Port = 3001; p = 0.47, Shannon-Wiener: Reef = 7.26 vs Port = 7.20; p = 0.45, and Inverse Simpson: Reef = 654.24 vs Port = 584.8; p = 0.74) (Fig 3A). In contrast, within-group (permutest, p = 0.023) and between-group beta-diversity were significantly different (PERMANOVA, Fmodel = 6.127, $R^2$ = 0.55 p = 0.03, Fig 3B), with the microbial communities in Reef sediments having a more dispersed beta-diversity compared to Port sediments (average distance to centroid: Reef = 55.76 vs Port = 41.99). Of the seven most relatively abundant bacteria orders, the Reef and Port sediments shared four taxa (57%) with NB1-J and Cyanobacteriales being unique to Reef sediments, and Desulfobacterales being unique to the Port sediments (Fig 3C).

Differential abundance analysis was conducted to identify the differences between sites, resulting in 293 significant ASVs (Fig 4). Of these, there were a total of 74 orders, 82 families,

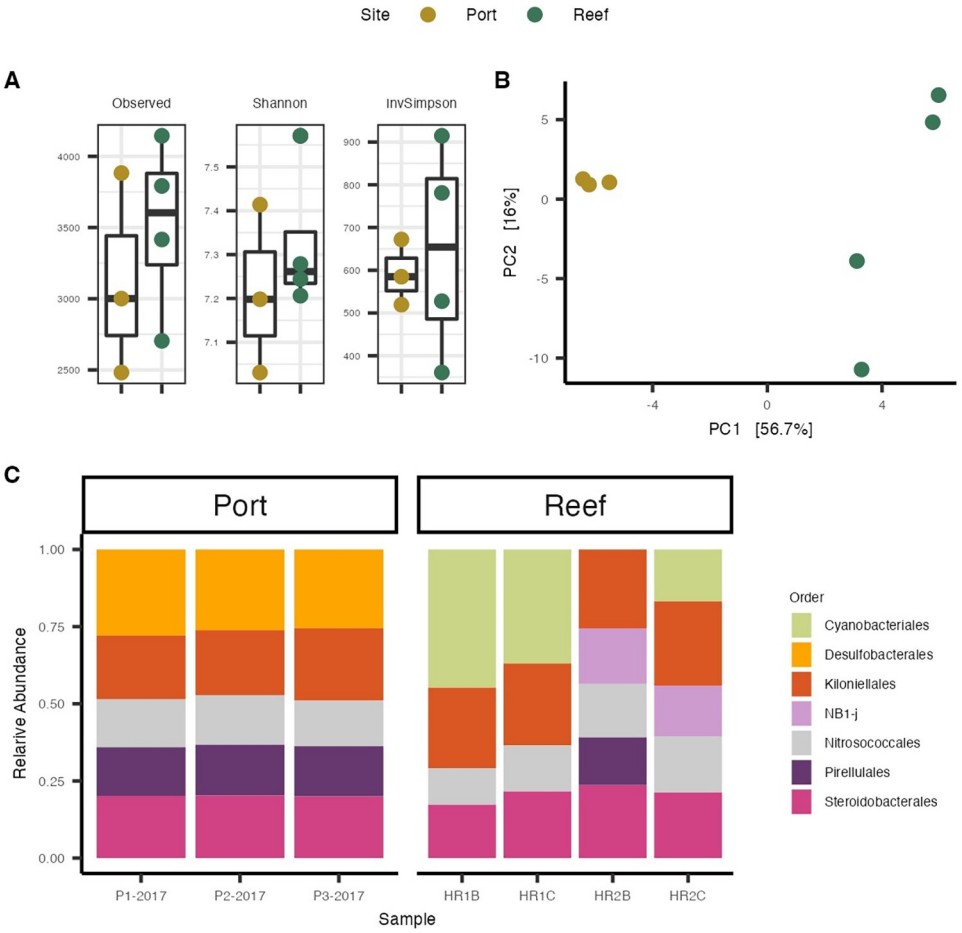

**Fig 3. Microbial communities found in Port and Reef sediments.** (A) Alpha diversity differences between Reef and Port samples using three diversity metrics: (1) Species richness (2) Shannon diversity Index, and (3) Inverse Simpson Index. (B) Beta diversity of microbial composition between Port and Reef sediment samples. (C) The cumulative relative abundances of the most abundant (0.05>) microbial order per sample grouped by the site of sediment collections.

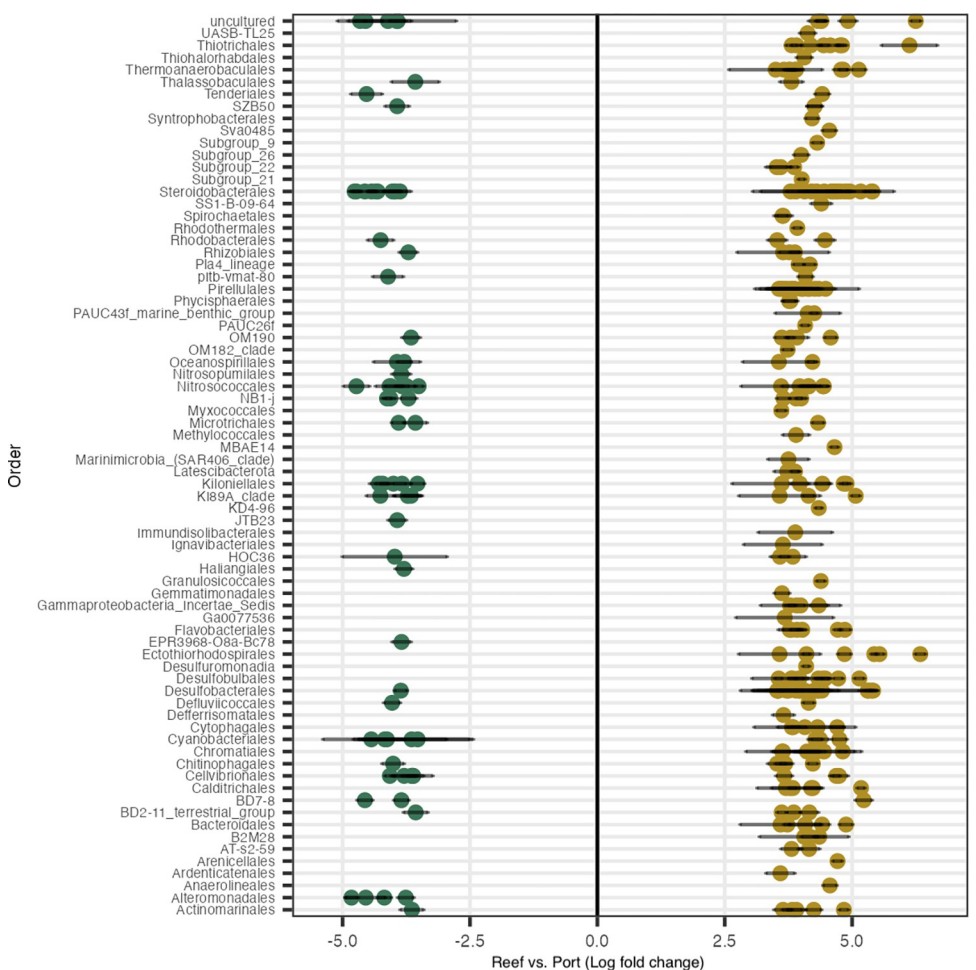

**Fig 4. Microbial Amplicon Sequence Variants (ASVs) associated with reef or Port sediments.** Differential abundances between Reef and Port, where the y-axis shows significant ASVs (padj<0.001, W statistic >90) grouped by order. Only ASVs with a log-fold change <−3.5 and >3.5 were visualized.

and 96 genera. The most prevalent differentially abundant orders were Steroidobacterales (N = 29), Desulfobacterales (N = 19), and Pirellulales (N = 15). The Port sediments had more enriched ASVs (N = 223) compared to Reef sediments (N = 70). More specifically, the ASVs most enriched in Port sediments were from the orders Ectothiorhodospirales (family *Ectothiorhodospiraceae*, genus *Thiogranum*), an uncultured genus from the Class Gammaproteobacteria, Desulfobacterales (*Desulfosarcinaceae*, Sva0081 sediment group), and Steroidobacterales (*Woeseiaceae*, *Woeseia*). In the Reef sediments, the most enriched orders were Alteromonadales (*Moritellaceae* from an uncultured genus), Steroidobacterales *(Woeseiaceae*, *Woeseia*), and Nitrosococcales (*Nitrosococcaceae*, *AqS1*).

## Discussion

In this study, we investigated differences in the survival and settlement of larvae from the threatened Caribbean coral *Orbicella faveolata* after short-term exposure to suspended sediments collected near the recently dredged Port of Miami channel (considered anthropogenically modified) versus sediments collected from the natal reef of the parent colonies in the Florida Keys (considered relatively natural or unmodified). Overall, we found that only *O.*

*faveolata* larvae exposed to Port suspended sediments had significantly lower survival and settlement rates compared to the control treatment (Fig 2). Though the high-dose Reef sediment treatments were associated with somewhat lower larval performance, these were not significantly different from the control treatment. Further, the low-dose Reef sediment treatment was not detrimental to coral larvae, and showed similar outcome odds of survival and settlement compared to the no sediment (control) treatment (see S3 Fig).

Port sediments had different microbiome profiles and significantly higher abundances of potentially pathogenic bacteria compared to Reef sediments, which may explain the reduced settlement observed even in the low-dose treatment. We speculate that Port sediments could have been more detrimental to larvae compared to reef sediments due to differences in their physical and chemical composition (e.g., grain size and sediment composition), and other biological differences (e.g., different microbial communities). Together, these findings suggest that, although suspended sediments may be generally detrimental to the survival and recruitment potential of *O. faveolata* larvae, anthropogenic sediments, such as those occurring adjacent to recent dredging projects, may incur greater harm compared to natural reef sediments.

## Effects on larval survival

Only the high-dose Port sediment treatment significantly impacted the survival of *O. faveolata* larvae. However, exposure to both Port sediment treatments, as well as the high-dose Reef treatment also resulted in reduced survival proportions compared to the control and low-dose Reef sediment treatment (Fig 2A). The few studies conducted to date looking at the effects of suspended sediments in Pacific species suggest that coral larvae may be generally resistant to brief exposures of elevated suspended sediment concentrations, at least to 50 mg/L (*Acropora digitifera*, [25]), 100 mg/L (*Acropora tenuis*, [26]), and as high as 800 mg/L ([43]) for *Acropora millepora*, *A. tenuis*, and *Pocillopora acuta*. The latter study, [43], hypothesized that coral larvae might protect themselves by secreting mucus to help with sediment clearing. A recent study, [47], observed *O. faveolata* adult fragments constantly producing mucus to help with sediment clearing after different exposure periods to loads between 50–150 mg/L. However, in our study, we did not see any signs of mucus production by *O. faveolata* larvae during or after the exposure period. Our exposure period (24 h) was shorter than that the 60 h used in [43] so it is possible that it takes a longer exposure period for *O. faveolata* larvae to display this protective mechanism. In addition, the lack of this protective mechanism in *O. faveolata* larvae could explain why we observed significantly lower survival at lower suspended sediment concentrations compared to [43].

Suspended sediments may also reduce larval survival through physical abrasion [25] and reduced light over prolonged periods can also lead to decreased photosynthetic efficiency of larval symbionts and subsequent larval mortality from starvation [23]. However, in our study, we think it is unlikely that decreased light was the driver of our results, given the use of aposymbiotic (symbiont-free) larvae. Instead, we speculate that increased mortality of *O. faveolata* larvae in the high-dose Port sediment treatment could have been due to chemical or microbial differences in the sediment composition compared to the Reef sediments. A recent study [34] assessed the effects of sedimentation on recruits from the brooding coral *Porites astreoides* using either reef sediments collected from the site of adult colonies, or sediments collected near Port Everglades (Fort Lauderdale, Florida) to resemble the grain size composition often found in dredging areas. Similar to our study, the authors found that reef sediments did not negatively affect coral survival, but Port sediments did. However, since there were only trace amounts of mud-sized particles in our Port sediments (approximately 1.25%), we hypothesize that Port sediments may be more likely to have affected coral larval survival compared to the Reef sediments due to microbial processes rather than grain size differences.

**Effects on larval settlement**

Overall, we found significant effects of both Port sediment treatments after one week of sediment exposure, resulting in approximately 50–60% reduction in settlement in *O. faveolata* larvae compared to the control treatment (Figs 2B and S3). Though the control treatment average settlement may appear low (~20%), this is within the average range of settlement observed across multiple years for *O. faveolata* larvae in previous 24 h competency assays (see [70]), using reef rubble similar to this study.

Previous work looking at the effects of suspended sediments on coral settlement has yielded mixed results, likely due to the different sediments used or methodological differences. Suspended sediment concentrations as low as ~50 mg/L reduced coral larval settlement for the Pacific coral species *A. digitifera* ([25]; reviewed in [23]), which the authors hypothesized could have been due to sediments which accumulated in the surface of the settlement plates. Ricardo et al. [43], however, did not find significant effects of suspended sediment concentrations in larval settlement of three Pacific coral species despite exposing the embryos to concentrations up to 1000 mg/L. In this case, the authors hypothesized that mucus secretion by ciliated larvae may have aided with attachment to the substrate. Then, a more recent study ([26]), found latent negative effects on the settlement success of Pacific coral *A. tenuis* larvae that had developed from embryos exposed to elevated suspended sediments, but not in the 3- or 5-day old larvae whose embryos were not pre-exposed. The authors thus suggested that embryo development might be a more sensitive stage to the effects of suspended sediments compared to larvae.

Surprisingly, we showed significant effects on larval settlement at 20 mg/L Port sediment concentrations; lower than those reported by [25] and other studies. We also tested potential latent effects of sediment exposure on eventual settlement one week after acute exposure. Thus, the settlement environment (i.e., light and condition or exposure of settlement substrates) were the same across treatments, and consequently, neither sediment deposition on the coral surfaces or settlement substrates, nor changes in light, were drivers of the reduced settlement observed in the exposed larvae. Furthermore, since neither of our Reef sediment treatments resulted in significant detrimental effects to coral settlement (but both the Port sediment treatments did), we speculate that biological/chemical damage to larvae exposed to Port sediments may have disrupted the detection of settlement cues, for example, via potential interactions with any contaminants or bacterial pathogens present in these sediments. Sediment-stressed larvae could also have had lower energy reserves available for allocation to settlement and subsequent metamorphosis, although no significant respiration rate differences were observed among treatments (see S4 Fig). Further, we believe that one of the reasons why we may have observed strong latent settlement effects in larvae exposed to Port sediments is that we used sediments that had not been dried or sterilized prior to exposure (but see [26]). Oven drying can change the chemical composition of the sediments and kill bacteria [5], masking these potentially important biological or chemical effects. The inclusion of "wet" sediments in our study may therefore have helped preserve these properties and allowed us to explore how bacterial assemblages can affect larval survival and settlement.

**Microbiome analysis of sediment samples and potential implications for coral survival and settlement**

Coral settlement is the result of complex behavioral responses that are modulated by distinct microbial signatures in CCA [39–42]. In this study, we hypothesized that microbial differences among the two sediment types may have resulted in decreases in survivorship and potential impairment of the ability of larvae to detect cues for settlement and metamorphosis. The

microbiome analysis of the sediment samples collected near the Port revealed differences in microbial beta-diversity and differentially abundant bacteria taxa compared to Reef sediments (Figs 3B, 3C and 4). Relative abundance analysis of the most abundant taxa showed that Desulfobacterales were unique to the Port sediments (Fig 3C). These taxa have been enriched in microbial communities from other inlets in southeast Florida [71] and have also been associated with black band disease [72], suggesting a role in coral health. Moreover, Desulfobacterales are anaerobic sulfate-reducing bacteria [73] that reduce sulfate ($SO_4^{2-}$) to hydrogen sulfide ($H_2S$). Desulfobacterales are abundant in organic-rich and high nutrient environments linked to eutrophication [74], and their waste product, $H_2S$, is correlated with coral tissue degradation [11], and algal mortality [75]. Port sediments enriched in Desulfobacterales (Fig 4) may therefore reflect elevated $H_2S$ and stressful hypoxic conditions that may have resulted in lower larval survival and settlement rates (Fig 2).

Sediments may also act as vectors for the transmission of diseases such as Stony Coral Tissue Loss Disease [22], first reported in outbreak levels (>5% prevalence) in waters of Miami-Dade County around September 2014 ([76,77] but see [78,79]), while the Port of Miami expansion dredging project was underway. Also, bacteria genera from two common human pathogens, *Staphylococcus* and *Streptococcus*, have been found across multiple time points in surface water and sediment samples near Port Everglades [71,80], another southeast Florida inlet subject to dredging impacts. Our findings add to these studies and also suggest that port inlets may carry potentially pathogenic bacteria which may be transported to nearby reefs through various sediment transport processes (natural or anthropogenic), resulting in potential exposure to coral larvae and other downstream health impacts.

## Effects on coral larval respiration

Despite significant effects of suspended sediments on *O. faveolata* larval survival and settlement (Figs 2 and S3), none of the sediment treatments had significant effects on coral larval respiration (S4 Fig and S4 Table). Sedimentation has been shown to decrease photosynthesis to respiration (P/R) ratios in adult corals, via decreased photosynthesis of the algal symbionts and increased respiration of the host (e.g., [81]). Increased mucus production (as a result of elevated turbidity or sedimentation) has also been linked to higher cellular respiration rates [82,83]. However, in this study, we used aposymbiotic larvae and did not see evidence of mucus production, which may explain the lack of significant effects on respiration. Alternatively, changes in respiration may have not been detected because it is not possible to maintain the sediment treatments in the microplate reader while respiration measurements are taken. Lower respiration rates across all treatments at the 1-week recovery period (compared to the 24-hr sediment exposure, see S3 Fig) could have resulted from ontogenetic and/or behavioral differences during those two time points (active swimming versus exploration of suitable substrate for settlement), e.g. [84]).

## Management implications for the conservation of corals

Our study is the first to examine the effects of suspended sediments on larvae from a Caribbean coral species. With the two major commercial shipping ports in southeast Florida planning major expansions that require multi-year dredge projects, this information is important for understanding the impacts of sedimentation on vulnerable life stages of corals, and enabling the development of protective management strategies. *O. faveolata* is listed under the US Endangered Species Act (ESA) and already shows negligible recruitment [49,85,86]. Recent work confirms the ESA Critical Habitat (legal) requirements that larvae require substrates free of sediment for settlement [33]. Importantly, our study showed that even brief exposures of *O.*

*faveolata* larvae to low doses of anthropogenic suspended sediments (such as those found near the recently dredged Port of Miami) during the pelagic phase could reduce larval survival and disrupt settlement, potentially compromising the recruitment success and recovery potential of this threatened species. Furthermore, we showed substantial direct mortality from short-term sediment exposure, but also an additional significant latent effect on the surviving larvae's capacity to successfully settle one week after sediment exposure. These results indicate that sediments which remain adjacent to a dredging project even after ~2 years may still have significant effects on early life stages of corals during relatively short-term resuspension events (e.g., storms). This highlights the potential for long-term impacts of dredging (e.g., critical habitat modification) as a result of resuspension of anthropogenic sediments. However, it is unclear how long such sediments can persist in the area, or how often resuspension events might happen.

Our findings also highlight the importance of setting temporary moratoriums as a management tool during dredging operations. Since 1993 dredging projects in Western Australia conducted near coral reefs have been required to implement a coral environmental ("stand-down") window spanning five days before spawning to seven days after spawning [13,87]. The *O. faveolata* larvae used in our experiments were eight days old and were adversely impacted by Port sediments with just 24 h of exposure, with detrimental latent effects still detectable after one week. Consequently, a one-week "stand down" period may be insufficient for broadcast spawner coral species such as *O. faveolata* given the length of time it takes for larvae to become competent [70] and find suitable substrate for settlement. Based on our results, we recommend setting environmental windows which encompass the months of peak spawning (typically August and September for *O. faveolata* [88,89]), plus an additional period of at least two to three weeks after spawning to prevent negative effects on settlement. These months are also typically the peak period for temperature-induced coral bleaching, and avoiding dredging during these months would therefore have multiple benefits.

Finally, we showed that sediments collected near the Port contained higher abundances of potentially pathogenic bacteria compared to Reef sediments, aligning with previous work conducted in the vicinity of the Port Everglades inlet [71,80]. These studies highlight the importance of monitoring for potential contaminants in sediments and changes in the microbiome which may result from dredging. In addition, although we only looked at the effects of suspended sediments on coral larval performance, suspended sediments can act in concert with other stressors typically associated with dredging. For example, [53] showed that elevated suspended sediment concentrations, when simultaneously combined with reduced light levels, resulted in partial mortality for multiple Pacific coral species. These findings emphasize the importance of monitoring both suspended sediments and benthic light availability (PAR) during large-scale dredging projects, and highlight the need for similar studies for Atlantic coral species.

## Supporting information

**S1 Fig. Composite display of grain size analysis of sediment samples.** Two replicates per sediment type (Port versus Reef) were used.
(DOCX)

**S2 Fig. Example plot of oxygen consumption over time for 6 *Orbicella faveolata* larvae placed in a single well of the microplate.** The $R^2$ value is given for the linear trend. The first 10–20% (pO2 in % air saturation) linear decreases in oxygen were used to calculate respiration rates per larva. Any portion of the slope that dipped below 70% air saturation was not used for data analysis.
(DOCX)

**S3 Fig. Odds ratios (OR) for the association between *Orbicella faveolata* larvae exposure to sediment treatments.** (A) larval survival after 24 h of treatments, and (B) larval settlement after one week of recovery from experimental treatments. OR = 1 Exposure does not affect odds of outcome, OR>1 Exposure associated with higher odds of outcome, and OR<1 Exposure associated with lower odds of outcome.
(DOCX)

**S4 Fig. Respiration rates (mean ± 95 CI) in *Orbicella faveolata* larvae** after a 24 h exposure to experimental treatments (left panel) and after one week of recovery from experimental treatments (right panel). White dots denote individual replicate wells.
(DOCX)

**S1 Table. Size fraction analysis of sediment samples.** Values correspond to graphic in S1 Fig.
(DOCX)

**S2 Table. Generalized linear mixed-effect models outputs for *Orbicella faveolata* larvae performance.** npar: Number of parameters, AIC: Akaike information criterion BIC: Bayesian information criterion, logLik: Log likelihood.
(DOCX)

**S3 Table. Final generalized linear mixed-effect model outputs for *Orbicella faveolata* performance.** (A) larval survival after 24 h of treatments, and (B) larval settlement after one week of recovery from experimental treatments. The coefficient estimates describe the change in the log odds for each treatment level compared to the base level (control).
(DOCX)

**S4 Table.** (A) Linear mixed-effect model outputs for *Orbicella faveolata* larvae respiration. (B) Pairwise comparisons between significant factors. DF: Degrees of freedom, CL: Confidence limit.
(DOCX)

## Acknowledgments

The authors thank C. Sinigalliano, B. Jensen, J. Destache, C. Pasparakis, and B. Young for assistance in the laboratory. Field assistance was provided by scientists from the NOAA Southeast Fisheries Science Center (D. Williams, A. Bright and A. Peterson). The authors also thank J. Hendee for his strong support and mentorship. J. Wolfe provided GIS assistance with Fig 1. *O. faveolata* gametes were collected under permit FKNMS-2016-047-A1 to M.W. Miller. Sediment collections were determined to be exempt from requirements for an Environmental Resource Permit (ERP) by the Florida's Department of Environmental Protection (DeMinimus exemption). This research was carried out in part under the auspices of the Cooperative Institute for Marine and Atmospheric Studies (CIMAS), a cooperative institute of the University of Miami and the National Oceanic and Atmospheric Administration (NOAA), Cooperative Agreement NA 20OAR4320472. The scientific results and conclusions, as well as any opinions expressed herein, are those of the author(s) and do not necessarily reflect the views of NOAA or the Department of Commerce.

## Author Contributions

**Conceptualization:** Xaymara M. Serrano, Stephanie M. Rosales, Margaret W. Miller, Ana M. Palacio-Castro, Andrew C. Baker.

**Data curation:** Xaymara M. Serrano, Stephanie M. Rosales, Ana M. Palacio-Castro.

**Formal analysis:** Xaymara M. Serrano, Stephanie M. Rosales, Ana M. Palacio-Castro.

**Funding acquisition:** Xaymara M. Serrano, Margaret W. Miller, Andrew C. Baker.

**Investigation:** Xaymara M. Serrano, Stephanie M. Rosales, Margaret W. Miller, Ana M. Palacio-Castro, Olivia M. Williamson, Andrea Gomez.

**Methodology:** Xaymara M. Serrano, Stephanie M. Rosales, Margaret W. Miller, Ana M. Palacio-Castro.

**Project administration:** Xaymara M. Serrano, Stephanie M. Rosales, Margaret W. Miller, Andrew C. Baker.

**Resources:** Xaymara M. Serrano, Stephanie M. Rosales, Margaret W. Miller, Ana M. Palacio-Castro, Andrew C. Baker.

**Software:** Stephanie M. Rosales, Ana M. Palacio-Castro.

**Supervision:** Xaymara M. Serrano, Margaret W. Miller, Andrew C. Baker.

**Validation:** Xaymara M. Serrano, Stephanie M. Rosales, Ana M. Palacio-Castro.

**Visualization:** Xaymara M. Serrano, Stephanie M. Rosales, Ana M. Palacio-Castro.

**Writing – original draft:** Xaymara M. Serrano, Stephanie M. Rosales, Ana M. Palacio-Castro.

**Writing – review & editing:** Xaymara M. Serrano, Stephanie M. Rosales, Margaret W. Miller, Ana M. Palacio-Castro, Olivia M. Williamson, Andrea Gomez, Andrew C. Baker.

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
