## [Decision Letter · Decision Letter 0]

1 Feb 2024

PONE-D-23-30672Sediment source and dose influence the larval performance of the threatened coral Orbicella faveolataPLOS ONE

Dear Dr. Serrano,

Thank you for submitting your manuscript to PLOS ONE. After careful consideration, we feel that it has merit but does not fully meet PLOS ONE’s publication criteria as it currently stands. Therefore, we invite you to submit a revised version of the manuscript that addresses the points raised during the review process.

We look forward to receiving your revised manuscript.

Kind regards,

Satheesh Sathianeson, Ph.D

Academic Editor

PLOS ONE

Journal Requirements:

2. We note that Figure 1 in your submission contain map/satellite images which may be copyrighted. All PLOS content is published under the Creative Commons Attribution License (CC BY 4.0), which means that the manuscript, images, and Supporting Information files will be freely available online, and any third party is permitted to access, download, copy, distribute, and use these materials in any way, even commercially, with proper attribution. For these reasons, we cannot publish previously copyrighted maps or satellite images created using proprietary data, such as Google software (Google Maps, Street View, and Earth). For more information, see our copyright guidelines: http://journals.plos.org/plosone/s/licenses-and-copyright.

Reviewers' comments:

Reviewer's Responses to Questions

**Comments to the Author**

1. Is the manuscript technically sound, and do the data support the conclusions?

Reviewer #1: Partly

Reviewer #2: Partly

2. Has the statistical analysis been performed appropriately and rigorously? 

Reviewer #1: Yes

Reviewer #2: Yes

3. Have the authors made all data underlying the findings in their manuscript fully available?

Reviewer #1: Yes

Reviewer #2: Yes

4. Is the manuscript presented in an intelligible fashion and written in standard English?

Reviewer #1: Yes

Reviewer #2: Yes

5. Review Comments to the Author

Reviewer #1: Serrano et al present a nice study in a well written paper, with interesting results. The strong effect of suspended sediments, particularly from Port Miami, on coral larval performance has wide reaching implications and will make an important contribution to the literature. Before acceptance I have a couple of potentially major comments that need to be responded to, along with a few minor changes recommended.

Major comments:

My main comment is in relation to the sediments used and their similarity, particularly looking at Fig 1D:E. The premise of the paper is that the suspension/resuspension of sediments may negatively affect larval performance, but if Fig 1D and E are good representations of the sediments used then they suggest that reef sediments are primarily sand based (E looks like a sand patch) and port sediment laden turf algae (D looks like sediment laden turf on top of hardbottom). This would also explain much of the difference between sediment composition. Both these sediment types are likely present in both locations but you’d expect different responses of larval performance and microbial community to each (coarser sand or finer sediment). It may just be that a more appropriate image is used which shows the sediments used, but if not some justification of why sediment from equivalent habitats (i.e., both sediment laden turf on hardbottom, or both sand patches) was not used.

I realise the aim of the study is to assess the impact of dredging activity and resuspension, but some information regarding suspended sediment concentrations expected on natural reefs would be very helpful, particularly as even the low doses (20 mg/L) would be considered high.

I also have some fairly minor comments with regard to the statistical analysis that need to be addressed so suitability of the results is clearer.

Other comments and suggestions:

It needs to be clear throughout that the study assesses the impact of suspended sediments (this isn’t mentioned in the abstract for instance).

Abstract

L29 – suggest noting exposed to suspended sediments

Introduction

L74- 84 - I find this paragraph a bit misleading in relation to the experiment as the influence of benthic sediment on settlement/survival isn't what's assessed. The information here is fine, but it would benefit from being clear that most of the focus of the affect of sedimentation on recruitment focuses on the benthos and not suspended sediments.

Methods

Suggest moving section on respiration in the methods before the settlement assay section so that the experimental design is chronological.

L213 – how long were respiration experiment runs?

L235 – this is a GLMM not a GLM as you’re using a random effect

L242 - Was model selection performed to assess whether both factors have a significant effect?

L243 - How did you assess post-hoc differences or did you just compare to control for base GLM summary? How were models validated to assure assumptions met?

L262 - Why did you use euclidean distance to calculate dissimilarity matrix rather than methods more suitable for biotic community data such as Bray-Curtis?

Results

L273 – 278 - I suggest sticking to the results of the experiments and the statistical tests performed here rather than the semi-qualitative descriptions used, particularly if you’re stating significant impact - Please provide detail on analysis - at minimum the test performed and the p value.

Fig 2. Caption – please note test used for significance

L305 – please note test significance from

Fig 3 - Should the title in 3A plot 1 be Richness not observed?

Discussion

L359 – 361 - This differs to what Fig2 says, please check this statement is correct

L371-372 - This doesn't seem to have been statistically tested in the results and Fig2 suggests no difference between high port and high reef survivorship

L387 – 390 - Similarly to in the intro, I don't understand why sediment covering on settlement surfaces is discussed as if it may have influenced the results. Maybe I've missed something here but the settlement assays didn't include any sediment but used filtered seawater, so sediment on surfaces is not what's affecting settlement? I think it's interesting enough that you find such strong effects just from suspended sediment

L432 – SCTLD first appearing at Port Miami late isn't entirely true ... see Jones et al 2021 (https://doi.org/10.1038/s41598-021-93111-0) or Hayes et al 2022 (doi: 10.3389/fmars.2022.975894) for suggestion that SCTLD was present prior to this.

Reviewer #2: This paper delves into the issue surrounding port development and dredging near Miami, which is a topical and controversial area of research. Predicting how corals may respond in such environments that are naturally already turbid is crucial to understanding the risk associated with these operations. The study investigates effects of suspended sediments on larvae, and latency effects of this exposure on settlement. Sediments-stress experiments are complicated to undertake because no two sediments are the same, and several mechanisms may be at play.

This study did an adequate job at sediment characterisation and measuring the biological responses. Overall, I suggest accepting the manuscript with major revisions. I write ‘major’ because I believe the data analysis needs to be checked, and the results altered to reflect this if changes are made. However, these changes should be reasonably quick to make.

The effect sizes here (differences from the control) for larval survivorship are quite modest, even at elevated SSCs. Personally, I would interpret this as larvae survival being initially quite robust to suspended sediment exposure. The settlement data however, shows stronger effects (>50% decrease) indicates a much greater concern despite some uncertainty.

Some feedback for future studies would be:

- Use a greater number treatment levels so that ‘effect thresholds’ are derived rather than NOEC/LOEC thresholds, which are produced from categorical analyses (Warne & van Dam 2008).

- Better experimental justification of the treatment levels. Surprisingly, the authors justify the suspended sediment concentrations from Australian dredging operations, rather than those from the Port of Miami. A quick scan of the grey literature and I was able to find government reports on suspended sediments in the region.

- More testing of the sediment suspending apparatus would have been useful. At the grain size reported, it is likely lots of agitation would be needed to maintain these in suspension and some may have settled out. i.e., responses could be worse than reported because the actual SSC might be lower than what was initially added.

Strengths

- For the most part, the experiment has a large sample size, with plenty of replication.

- The paper was written very clearly and was easy to follow. Thank you for that.

- The microbial sediment analysis supplemented the study well, this is not often done and can point to some mechanisms.

Weaknesses

- The study missed two crucial papers on the topic that closely matched the design used here (Ricardo et al. 2016, Humanes et al. 2017). Hopefully this was just an oversight. These studies found no real effects survivorship on larvae following sediment exposure, but some latency effects on settlement at very high SSCs in one study.

- Thank you for providing the data and code. However, even though the approach appears mostly sound, it seems there may be an issue in the statistical analyses for the survival and settlement data. The plots indicate 3 treatment levels are significant but it is clear that this is not the case by looking at the overlapping 95% CI. Reanalysis of the survival data only points to 1 treatment (‘High Port’) level being ‘significant’ compared to the control. This issue may be caused by how observational-level random effects are implemented, and which posthoc test is used (Harrison 2015).

Survival.data1$obs <- factor(seq_len(nrow(Survival.data1))) # unique container/observation ID

fit1<-glmer(cbind(Swimming, Dead) ~Treatment +(1|obs),

data=Survival.data1,family="binomial")

library(emmeans)

contrast(emmeans (fit1, ~ Treatment), "trt.vs.ctrl1", ref = 'Control')

- please write the exact p-values rather than <0.05 (unless <0.001)

Minor points -

- 108 Gamete ‘egg-sperm’

- 117 – How were the larvae transported?

- 143 - How long was this? Was there any water exchange?

- 144 Can you explain the wet weigh to dry weight procedure a bit more.

- 235 generalized linear ‘mixed ‘models (‘GLMM’)

- 302 – Please explain which sequencing filtering steps were taken.

- 341-342 This sentence is probably better suited to the Methods

- I was initially concerned that conducting a settlement assay on a random piece of rubble would yield no settlement. Corals are selective about settlement substrates and O. faveolata has been shown to show preferences to types of CCA. So I relieved to see settlement at ~20% in the control, which is typical of this species, but much lower than many Indo-pacific corals. That said, I think it is worthwhile in the Discussion or the Methods outlining that O. faveutous typically shows very low settlement success and providing a few references. As an aside, I do wonder if the preferred settlement substrate for O. faveolata has not been found, or if the settlement peak is very protracted.

- One point to consider is that even though 100 mg/L is considered very high (it is closest to the 100% percentile in US EPA MOMS report), larvae move passively within a plume and therefore might have a different exposure time to an in situ nephelometers from where most WQ data come from.

- 368 – Did you notice any mucous production from the larvae from clearing?

- 386 – This reference/hypothesis is unclear. Please cite the actual study rather than the literature review

- - 393. There have been other studies that have done this (see earlier).

- 406. Possibly, but Humanes 2017 uses wet sediments and did not find latency effects (although they did with pre-exposed embryos).

- 400. ‘we speculate that biological/chemical damage to larvae exposed to Port sediments may have disrupted the detection of settlement cues’ is maybe clearer.

Harrison XA (2015) A comparison of observation-level random effect and Beta-Binomial models for modelling overdispersion in Binomial data in ecology & evolution. PeerJ 3:e1114

Humanes A, Ricardo G, Willis B, Fabricius K, Negri A (2017) Cumulative effects of suspended sediments, organic nutrients and temperature stress on early life history stages of the coral Acropora tenuis. Scientific Reports 7:44101

Ricardo GF, Jones RJ, Clode PL, Negri AP (2016) Mucous secretion and cilia beating defend developing coral larvae from suspended sediments. PloS ONE 11:e0162743

Warne MSJ, van Dam R (2008) NOEC and LOEC data should no longer be generated or used. Australas J Ecotoxicol 14:1

-

6. PLOS authors have the option to publish the peer review history of their article (what does this mean?). If published, this will include your full peer review and any attached files.

Reviewer #1: **Yes: **

Reviewer #2: No

---

## [Author Response · Author response to Decision Letter 0]

2 May 2024

Response to reviews

Manuscript ID PONE-D-23-30672

Sediment source and dose influence the larval performance of the threatened coral Orbicella faveolata

We would like to thank the Academic Editor and the two reviewers assigned for their constructive comments and suggestions; these greatly improved our manuscript. We have carefully evaluated all comments/suggestions and prepared a revised manuscript in the style and format of the journal PLOS ONE. 

Academic Editor comments

Comment 1: Please ensure that your manuscript meets PLOS ONE's style requirements, including those for file naming. The PLOS ONE style templates can be found at https://journals.plos.org/plosone/s/file?id=wjVg/PLOSOne_formatting_sample_main_body.pdf and 

Author’s response: We have prepared a revised manuscript and reviewed it to ensure that it meets PLOS ONE’s style requirements.

Comment 2: We note that Figure 1 in your submission contains map/satellite images which may be copyrighted. All PLOS content is published under the Creative Commons Attribution License (CC BY 4.0), which means that the manuscript, images, and Supporting Information files will be freely available online, and any third party is permitted to access, download, copy, distribute, and use these materials in anyway, even commercially, with proper attribution. For these reasons, we cannot publish previously copyrighted maps or satellite images created using proprietary data, such as Google software (Google Maps, Street View, and Earth). For more information, see our copyright guidelines: http://journals.plos.org/plosone/s/licenses-and-copyright.

We require you to either (a) present written permission from the copyright holder to publish these figures specifically under the CC BY4.0 license, or (b) remove the figures from your submission:

USGS National Map Viewer (public domain):

http://viewer.nationalmap.gov/viewer/

The Gateway to Astronaut Photography of Earth (public domain):

http://eol.jsc.nasa.gov/sseop/clickmap/

Maps at the CIA (public domain):

https://www.cia.gov/library/publications/the-world-factbook/index.html and

https://www.cia.gov/library/publications/cia-maps-publications/index.html

NASA Earth Observatory (public domain):

http://earthobservatory.nasa.gov/

Landsat:

http://landsat.visibleearth.nasa.gov/

USGS EROS (Earth Resources Observatory and Science (EROS) Center) (public domain):

http://eros.usgs.gov/#

Natural Earth (public domain):

http://www.naturalearthdata.com/

Author’s response: As suggested, figures 1A, 1B, and 1C have been revised using basemaps that comply with the CC BY 4.0 license. Figures 1A and 1C use the World Imagery Base Map from ESRI. Figure 1B uses the USGS ImageryOnly Base Map from the USGS National Map Viewer. The copyright information has now been included in the revised figure caption and the service layer credits are also displayed in each figure. In addition, we removed the words “Miller et al. 2016” from Figure 1C which may have erroneously given the impression that it was a copyrighted figure. Finally, the original figures 1D and 1E were photographs taken by one of the authors but these have been deleted to address one of Reviewer #1 concerns.

Comment 3: Note from Emily Chenette, Editor in Chief of PLOS ONE, and Iain Hrynaszkiewicz, Director of Open Research Solutions at PLOS: Did you know that depositing data in a repository is associated with up to a 25% citation advantage (https://doi.org/10.1371/journal.pone.0230416)? If you’ve not already done so, consider depositing your raw data in a repository to ensure your work is read, appreciated and cited by the largest possible audience. You’ll also earn an Accessible Data icon on your published paper if you deposit your data in any participating repository (https://plos.org/open-science/open-data/#accessible-data).

Author’s response: Thank you for the suggestion. All our data and code has been deposited in various repositories recommended by PLOS ONE. All the data and code for the larval survivorship, settlement and oxygen consumption analysis can be found in Zenodo at https://doi.org/10.5281/zenodo.8022069. Raw reads and metadata for the microbial 16S of sediment samples are available in NCBI’s SRA under BioProject ID PRJNA1008169. Finally, all data and code for the sediment microbial analysis can be found at https://github.com/srosales712/Larvae_Sediment.

Reviewer #1 summary

Serrano et al present a nice study in a well written paper, with interesting results. The strong effect of suspended sediments, particularly from Port Miami, on coral larval performance has wide reaching implications and will make an important contribution to the literature. Before acceptance, I have a couple of potentially major comments that need to be responded to, along with a few minor changes recommended.

Major comments

Comment 1: My main comment is in relation to the sediments used and their similarity, particularly looking at Fig 1D:E. The premise of the paper is that the suspension/resuspension of sediments may negatively affect larval performance, but if Fig 1D and E are good representations of the sediments used then they suggest that reef sediments are primarily sand based (E looks like a sand patch) and port sediment laden turf algae (D looks like sediment laden turf on top of hardbottom). This would also explain much of the difference between sediment composition. Both these sediment types are likely present in both locations but you’d expect different responses of larval performance and microbial community to each (coarser sand or finer sediment). It may just be that a more appropriate image is used which shows the sediments used, but if not some justification of why sediment from equivalent habitats (i.e., both sediment laden turf on hardbottom, or both sand patches) was not used.

Author’s response: To clarify, the aim of our study was to use sediments representative from these two different sites (Port vs. Reef) to use in our laboratory experiments, not sediments from equivalent habitats within these sites. As stated in the methods section, 3-4 containers with surficial sediments were collected per site and combined to produce a single “batch” for experiments. Thus, the pictures do not show the only area where sediments were collected from at each site. Since we do not have actual pictures of all the areas where the sediments were collected from, for clarity, we have deleted the pictures from Figure 1. Also, note that the grain size analysis conducted for both types of sediment (Port and Reef) showed that these were primarily characterized as sand (>97.4%, see Table S1 and Figure S1), with only trace amounts of mud-sized particles in the Port sediments.

Comment 2: I realise the aim of the study is to assess the impact of dredging activity and resuspension, but some information regarding suspended sediment concentrations expected on natural reefs would be very helpful, particularly as even the low doses (20 mg/L) would be considered high.

Author’s response: We appreciate the suggestion. We have now added additional information to the manuscript regarding suspended sediment concentrations typically observed in natural reefs. From our literature search, our low-dose suspended sediment treatments appear to be about twice the concentrations typically observed in natural reefs in southeast Florida (~10 mg/L; Whitall and Bricker, 2021), but within the range observed in inshore reefs in Hawaii (Jokiel et al. 2014), and in turbid reefs in the Great Barrier Reef, which can often exceed 50 mg/L during natural wind-wave events (Larcombe et al. 1995; Cooper et al. 2008; Bessell-Browne et al. 2017). 

Minor comments

Comment 1: It needs to be clear throughout that the study assesses the impact of suspended sediments (this isn’t mentioned in the abstract for instance).

Author’s response: Thank you for your comment; we did not realize that we had not mentioned “suspended sediments” in the abstract. For clarity, we added “suspended sediments” in the abstract and at various other sections throughout the manuscript.

Comment 2: L29 – suggest noting exposed to suspended sediments

Author’s response: Done.

Comment 3: L74- 84 - I find this paragraph a bit misleading in relation to the experiment as the influence of benthic sediment on settlement/survival isn’t what's assessed. The information here is fine, but it would benefit from being clear that most of the focus of the effect of sedimentation on recruitment focuses on the benthos and not suspended sediments.

Author’s response: Agree. The paragraph was modified to include a short overview of the effects of suspended sediments on coral larval survival and settlement as well. 

Comment 4: Suggest moving section on respiration in the methods before the settlement assay section so that the experimental design is chronological.

Author’s response: Done.

Comment 5: L213 – how long were respiration experiment runs?

Author’s response: The respiration runs varied in length (depending on the larval oxygen consumption), but typically lasted approximately 3-4 hours (Figure S2 shows an example plot over time).

Comment 6: L235 – this is a GLMM not a GLM as you’re using a random effect

Author’s response: The sentence was changed to GLMM to specify that the generalized linear mixed model includes a random effect.

Comment 7: L242 - Was model selection performed to assess whether both factors have a significant effect?

Author’s response: Yes. We now specify in the text that “Model selection was performed with stepwise backward elimination of non-significant terms using the “step” function of the lmerTest R package”.

Comment 8: L243 - How did you assess post-hoc differences or did you just compare to control for base GLM summary? How were models validated to assure assumptions met?

Author’s response: Pairwise differences between the Control and each of the other four treatments were evaluated with the emmeans package with an alpha value of 0.05. This has been added to the manuscript. Treatment was not a significant factor in oxygen consumption, so we did not perform a post-hoc test in this model. Best binomial models were chosen based on the AIC and are now all shown in the S2 Table. The best linear model was chosen by backward elimination of non-significant terms using the “step” function. This was also added to the methods.

Comment 9: L262 - Why did you use euclidean distance to calculate dissimilarity matrix rather than methods more suitable for biotic community data such as Bray-Curtis?

Author’s response: We used Euclidean distance because we followed the Compositional Data Analysis (CoDa) method. Bray-Curtis methods transform the data into ratios or relative abundance. CoDA methods move away from this method and transform the data using Centered Log-Ratios (CLR), which are used for compositional data like our sequence data. CLR uses the geometric mean of read counts from all taxa within a sample as the reference or denominator. This method involves dividing all taxon read counts in a sample by this geometric mean, allowing for the comparison of log-fold changes in this ratio between different samples. In this paper, we used Euclidean distance to achieve an Aitchison distance (CLR transformed data on a Euclidean space), which is the recommended distance matrix for compositional data 

(https://www.ncbi.nlm.nih.gov/pmc/articles/PMC5695134/).

Comment 10: L273 – 278 - I suggest sticking to the results of the experiments and the statistical tests performed here rather than the semi-qualitative descriptions used, particularly if you’re stating significant impact - Please provide detail on analysis - at minimum the test performed and the p value.

Author’s response: As suggested, we removed the qualitative descriptions used in this paragraph and provided the test performed and the p-value.

Comment 11: Fig 2. Caption – please note test used for significance

Author’s response: Done. Test was added to the caption.

Comment 12: L305 – please note test significance from

Author’s response: It was an ANOVA but this difference was not significant between sites (Fig 3A). For clarity, we added the p-values to the text.

Comment 13: Fig 3 - Should the title in 3A plot 1 be Richness not observed?

Author’s response: We believe this is interchangeable (it doesn’t affect the meaning either way) and prefer to leave the figure caption as is.

Comment 14: L359 – 361 - This differs to what Fig 2 says, please check this statement is correct.

Author’s response: Thanks for pointing out this error. Based on recommendations from Reviewer #2 we have corrected the data for overdispersion which also changed these results. These changes have now been incorporated into the manuscript.

Comment 15: L371-372 - This doesn't seem to have been statistically tested in the results and Fig 2 suggests no difference between high port and high survivorship.

Author’s response: We believe the reviewer meant to write “no difference between high port and high reef survivorship”. Based on recommendations from Reviewer #2 we have corrected the data for overdispersion which also changed these results. These changes have now been incorporated into the manuscript and results from the statistical analysis incorporated in the Results section.

Comment 16: L387 – 390 - Similarly to in the intro, I don't understand why sediment covering on settlement surfaces is discussed as if it may have influenced the results. Maybe I've missed something here but the settlement assays didn't include any sediment but used filtered seawater, so sediment on surfaces is not what's affecting settlement? I think it's interesting enough that you find such strong effects just from suspended sediment.

Author’s response: Thank you for your comment. We realized this was an oversight. The introduction and discussion sections were modified to include a short overview of the effects of suspended sediments on coral larval survival and settlement. We also included a few relevant references in these sections as pertinent.

Comment 17: L432 – SCTLD first appearing at Port Miami late isn't entirely true ... see Jones et al 2021 (https://doi.org/10.1038/s41598-021-93111-0) or Hayes et al 2022

 (doi:10.3389/fmars.2022.975894) for suggestion that SCTLD was present prior to this.

Author’s response: 

---

## [Decision Letter · Decision Letter 1]

16 May 2024

PONE-D-23-30672R1Sediment source and dose influence the larval performance of the threatened coral Orbicella faveolataPLOS ONE

Dear Dr. Serrano,

Thank you for submitting your manuscript to PLOS ONE. After careful consideration, we feel that it has merit but does not fully meet PLOS ONE’s publication criteria as it currently stands. Therefore, we invite you to submit a revised version of the manuscript that addresses the points raised during the review process.

We look forward to receiving your revised manuscript.

Kind regards,

Satheesh Sathianeson, Ph.D

Academic Editor

PLOS ONE

Journal Requirements:

Reviewers' comments:

Reviewer's Responses to Questions

**Comments to the Author**

1. If the authors have adequately addressed your comments raised in a previous round of review and you feel that this manuscript is now acceptable for publication, you may indicate that here to bypass the “Comments to the Author” section, enter your conflict of interest statement in the “Confidential to Editor” section, and submit your "Accept" recommendation.

Reviewer #1: All comments have been addressed

Reviewer #2: All comments have been addressed

2. Is the manuscript technically sound, and do the data support the conclusions?

Reviewer #1: Yes

Reviewer #2: Yes

3. Has the statistical analysis been performed appropriately and rigorously? 

Reviewer #1: Yes

Reviewer #2: Yes

4. Have the authors made all data underlying the findings in their manuscript fully available?

Reviewer #1: Yes

Reviewer #2: Yes

5. Is the manuscript presented in an intelligible fashion and written in standard English?

Reviewer #1: Yes

Reviewer #2: Yes

6. Review Comments to the Author

Reviewer #1: Authors appropriately answered all comments and I recommend publishing the manuscript.

There are a couple of minor things I noticed that the authors may want to check:

One minor typo in table S2b - settlement should be the response in all model equations. Also, check p-values in text L297-308 - I presume the values stated in text are from the emmeans pairwise comparisons and not the glmm summary that's in table S3? I got confused at first because S3 table is referenced.

Reviewer #2: The authors have adequately addressed all of my comments and concerns and I look forward to seeing this manuscript published

7. PLOS authors have the option to publish the peer review history of their article (what does this mean?). If published, this will include your full peer review and any attached files.

Reviewer #1: **Yes: **

Reviewer #2: No

---

## [Author Response · Author response to Decision Letter 1]

26 May 2024

Response to reviews

Manuscript ID PONE-D-23-30672R1

Sediment source and dose influence the larval performance of the threatened coral Orbicella faveolata

We have carefully evaluated the minor comments and suggestions received and prepared a revised manuscript in the style and format of the journal PLOS ONE. 

Academic Editor comments

Comment 1: Please review your reference list to ensure that it is complete and correct. If you have cited papers that have been retracted, please include the rationale for doing so in the manuscript text, or remove these references and replace them with relevant current references. Any changes to the reference list should be mentioned in the rebuttal letter that accompanies your revised manuscript. If you need to cite a retracted article, indicate the article’s retracted status in the References list and also include a citation and full reference for the retraction notice.

Author’s response: We have reviewed our reference list to ensure that is complete and correct. We also used the style file (PLoS) offered by EndNote to assist with formatting our references. Note that no changes to the reference list were made in this revised version of the manuscript (PONE-D-23-30672R1). However, below we provide a list of all the references that were replaced, added or removed in our previous version of the manuscript and rationale for doing so. 

1) We removed one reference (Vazquez, 2023, citation below) and replaced it with the citation of the actual Port Miami report from NOAA referenced in the Local 10 News.

Vazquez C. Explosive report finds PortMiami dredging caused extensive coral reef damage. Local 10 News [Internet]. 2023 Sept 06. Available from: 

https://www.local10.com/news/local/2023/09/06/explosive-report-finds-portmiami-dredging-caused-extensive-coral-reef-damage/#:~:text=MIAMI%20%E2%80%93%20A%20new%20federal%20report,along%20the%20port's%20entrance%20channel.

Replaced with:

[45] Karazsia J, Mack K, Wilber P, Miller M, Griffin S, Moore T. Examination of Sedimentation Impacts to Coral Reef along the Port Miami Entrance Channel, December 2015 and April 2016. Available from: 

https://www.ncei.noaa.gov/data/oceans/coris/library/NOAA/CRCP/NMFS/SERO/Projects/30049/Karazsia2023_PortMiami_Dredge_PhaseIII_ImpactAssessment_FinalReport.pdf: 2023.

2) We added several references in our previous version of the manuscript to address the two reviewers’ concerns and suggestions. Most of these references were identified in our previous rebuttal letter and all references were shown as tracked changes. These references were incorporated in the R1 manuscript and correspond to the following citations:

[25] Gilmour J. Experimental investigation into the effects of suspended sediment on fertilisation, larval survival and settlement in a scleractinian coral. Marine Biology. 1999;135:451-62.

[26] Humanes A, Ricardo GF, Willis BL, Fabricius KE, Negri AP. Cumulative effects of suspended sediments, organic nutrients and temperature stress on early life history stages of the coral Acropora tenuis. Scientific Reports. 2017;7(1):44101.

[43] Ricardo GF, Jones RJ, Clode PL, Negri AP. Mucous secretion and cilia beating defend developing coral larvae from suspended sediments. PLoS One. 2016;11(9):e0162743.

[51] Whitall D, Bricker S. Examining Ambient Turbidity and Total Suspended Solids Data in South Florida Towards Development of Coral Specific Water Quality Criteria. 2021. doi:

doi.org/10.25923/v35e-cv79.

[52] Jokiel PL, Rodgers KuS, Storlazzi CD, Field ME, Lager CV, Lager D. Response of reef corals on a fringing reef flat to elevated suspended-sediment concentrations: Molokai, Hawaii. PeerJ. 2014;2:e699.

[54] Cooper TF, Ridd PV, Ulstrup KE, Humphrey C, Slivkoff M, Fabricius KE. Temporal dynamics in coral bioindicators for water quality on coastal coral reefs of the Great Barrier Reef. Marine and Freshwater Research. 2008;59(8):703-16.

[55] Larcombe P, Ridd P, Prytz A, Wilson B. Factors controlling suspended sediment on inner-shelf coral reefs, Townsville, Australia. Coral reefs. 1995;14:163-71.

[61] Harrison XA. A comparison of observation-level random effect and Beta-Binomial models for modelling overdispersion in Binomial data in ecology & evolution. PeerJ. 2015;3:e1114.

[63] Searle SR, Speed FM, Milliken GA. Population marginal means in the linear model: an alternative to least squares means. The American Statistician. 1980;34(4):216-21.

[70] Miller MW, Bright AJ, Pausch RE, Williams DE. Larval longevity and competency patterns of Caribbean reef-building corals. PeerJ. 2020;8:e9705.

[78] Jones NP, Kabay L, Semon Lunz K, Gilliam DS. Temperature stress and disease drives the extirpation of the threatened pillar coral, Dendrogyra cylindrus, in southeast Florida. Scientific Reports. 2021;11(1):14113.

[79] Hayes NK, Walton CJ, Gilliam DS. Tissue loss disease outbreak significantly alters the Southeast Florida stony coral assemblage. Frontiers in Marine Science. 2022;9. doi:

doi.org/10.3389/fmars.2022.975894.

[82] Riegl B, Branch GM. Effects of sediment on the energy budgets of four scleractinian (Bourne 1900) and five alcyonacean (Lamouroux 1816) corals. Journal of Experimental Marine Biology and Ecology. 1995;186(2):259-75.

 [83] Telesnicki G, Goldberg W. Effects of turbidity on the photosynthesis and respiration of two south Florida reef coral species. Oceanographic Literature Review. 1996;2(43):199.

3) Two references were removed from our previous version of the manuscript to help focus our introduction on the effects of suspended sediments on larval settlement and address the suggestions provided by reviewer #1. The citations for these references are below:

Price N. Habitat selection, facilitation, and biotic settlement cues affect distribution and performance of coral recruits in French Polynesia. Oecologia. 2010;163(3):747-58. Epub 2010/02/20. doi:

10.1007/s00442-010-1578-4. 

Ritson-Williams R, Paul VJ, Arnold SN, Steneck RS. Larval settlement preferences and post-settlement survival of the threatened Caribbean corals Acropora palmata and A. cervicornis. Coral Reefs. 2009;29(1):71-81. doi: 10.1007/s00338-009-0555-z.

Reviewer #1 summary

Authors appropriately answered all comments and I recommend publishing the manuscript. There are a couple of minor things I noticed that the authors may want to check.

Comment 1: One minor typo in table S2b - settlement should be the response in all model equations.

Author’s response: Thanks for noticing the error. We have corrected it.

Comment 2: Also, check p-values in text L297-308 - I presume the values stated in text are from the emmeans pairwise comparisons and not the glmm summary that’s in table S3? I got confused at first because S3 table is referenced.

Author’s response: Yes, that is correct. We have modified the text in lines 297-308 to indicate that the p-values refer to figure 2A and 2B and not the results from the glmm summary that is in table S3.

Reviewer #2 summary

The authors have adequately addressed all of my comments and concerns and I look forward to seeing this manuscript published.

---

## [Editor Report · Decision Letter 2]

28 May 2024

Sediment source and dose influence the larval performance of the threatened coral Orbicella faveolata

PONE-D-23-30672R2

Dear Dr. Serrano,

We’re pleased to inform you that your manuscript has been judged scientifically suitable for publication and will be formally accepted for publication once it meets all outstanding technical requirements.

Kind regards,

Satheesh Sathianeson, Ph.D

Academic Editor

PLOS ONE
---

## [Editor Report · Acceptance letter]

17 Jun 2024

PONE-D-23-30672R2 

PLOS ONE

Dear Dr. Serrano, 

I'm pleased to inform you that your manuscript has been deemed suitable for publication in PLOS ONE. Congratulations! Your manuscript is now being handed over to our production team.

Kind regards, 

on behalf of

Dr. Satheesh Sathianeson 

Academic Editor

PLOS ONE